# Assessing the Placement of the Cochlear Slim Perimodiolar Electrode Array by Trans Impedance Matrix Analysis: A Temporal Bone Study

**DOI:** 10.3390/jcm11143930

**Published:** 2022-07-06

**Authors:** Ángel Ramos de Miguel, Diego Riol Sancho, Juan Carlos Falcón-González, Joana Pavone, Leandro Rodríguez Herrera, Silvia Borkoski Barreiro, Nadia Falcón Benitez, Ángel Ramos Macias

**Affiliations:** 1Department of Otolaryngology, and Head and Neck Surgery, Complejo Hospitalario Universitario Insular Materno Infantil de Gran Canaria, 35016 Las Palmas, Spain; angel.ramos@fpct.ulpgc.es (Á.R.d.M.); jfalgond@gobiernodecanarias.org (J.C.F.-G.); drapavoneorl@gmail.com (J.P.); leo.rodriguez.h@gmail.com (L.R.H.); silviaborkoski@hotmail.com (S.B.B.); 2Department Otolaryngology, Psychoacoustic & Equilibrium Laboratory, University Institute of Intelligent Systems and Numeric Applications in Engineering, Las Palmas University (ULPGC), 35016 Las Palmas, Spain; 3Department of Radiology, Head Neck Surgery, Complejo Hospitalario Universitario Insular Materno Infantil de Gran Canaria, 35016 Las Palmas, Spain; diegoriolsancho@hotmail.com; 4Department of Clinical Sciences, University of Las Palmas de Gran Canaria (ULPGC), 35001 Las Palmas, Spain; nadiafalconb13@gmail.com

**Keywords:** cochlear implant, trans impedance, temporal bone, perimodiolar electrode array, hearing loss, fold-over of the electrode

## Abstract

New cochlear implant (CI) electrode arrays provide softer insertion dynamics; however, due to their high flexibility, the possibilities of fold-overs or intraoperative displacements must be taken into account. The position of each individual electrode can only be determined by using high-resolution computed tomography or cone-beam CT. The trans-impedance matrix test (TIM) is an electrophysiological method based on electric field imaging that can provide images of electrode position and electrode folding. Objective: In this experimental research, we evaluated the result of TIM as a method of monitoring cochlear insertion for a precurved slim modiolar electrode array in fresh human temporal bones by analyzing the transimpedance matrix patterns and their correlation with electrode position using high-resolution computed tomography. Material and Methods: Sixteen slim modiolar electrode arrays were inserted into eight fresh Human Temporal Bones. Eight electrodes were inserted according to the correct methodology of insertion, and eight were intentionally folded over. After all insertions, a trans-impedance matrix analysis and a Cone Beam CT (CBCT) were performed in each temporal bone. Results: If we correlated the TIM patterns with the radiological electrode position, we observed that better electrode intracochlear positions indicated more “homogeneous” TIM patterns (intracochlear voltage dropped monotonically as the distance between stimulation and recording contact increased, both toward the apex and toward the base). A correlation where fold-over was detected in the TIM results was found in all eight temporal bone radiological findings. Conclusions: Trans-Impedance Matrix patterns were correlated with the radiological CI electrode position. When a tip fold-over appeared, a matrix with a secondary ridge in addition to the primary ridge was observed in all cases. TIM can be an effective method in the control of electrode positioning.

## 1. Introduction

One of the most important steps in cochlear implant surgery is the insertion of the electrode array into the *scala tympani* (ST). Residual hearing preservation has been related to different surgical factors, such as the type and dimension of the array, the type of cochlear approach (cochleostomy or Round Window), the mechanism of insertion, and the use of lubricants or drugs in the cochlea [1,2].

Precurved electrode arrays are manufactured in a spiral configuration in an attempt to adjust to the modiolar area of the human cochlea. These electrode arrays were designed for intracochlear placement next to the modiolus. Perimodiolar electrodes have been designed and introduced to reduce energy consumption and increase the specificity of neuronal stimulation [3,4].

The cochlear diameter can vary from 7.0–9.0 mm [5,6]. There is also a significant correlation between the length and the width of the cochlear base. Not only the length, but also the number of cochlear turns is variable, ranging from 2.39 to 2.84 [5]. It is also known that in the apical direction, the ST becomes narrower, mainly at the external wall area; this can also be observed in the mid-modiolar area. While the height at the lateral wall decreases significantly after 450°, the modiolar height remains similar along the cochlear length. Between 450° and 500° is the point at which the first and second turns of the cochlea are close to each other and where the ST appears to be deflected upward. These sudden jumps may result in electrode arrays getting stuck at that location, and may lead to buckling of the electrode array or deviation from the ST trajectory and displacement toward the scala vestibuli [7].

Currently, the electrode array position within the cochlea can be determined by radiology imaging methods, for example, fluoroscopy [8], phase-contrast radiography [9], rotational-tomography (RT), or a combination of conventional radiography and computed tomographic (CT) images [10]. Conventional cochlear view (X-ray) or high-resolution CT (HRCT) are also commonly used for electrode insertion, scalar dislocations, or tip fold-over evaluations. The position of each individual electrode can only be determined by using high-resolution CT; however, this approach has the disadvantages of multiple artefacts introduced by metal elements and the extra radiation doses for the patient. Cone beam computed tomography (CBCT) has been validated as a valuable tool to assess post cochlear electrode implantations [11,12], with the use of less irradiation than a high-resolution CT and comparably less sensitivity to metal artefacts. It allows an easier identification of electrode placement in either the ST or the scala vestibuli (SV).

While radiological imaging is the standard method for the determination of the array position, the required equipment is not available in operating rooms in many medical centers, and so its use may be limited. The utility of these methods is also limited by the need for experts to perform analyses of the obtained images in order to confirm the correct position of the electrodes. In addition, the expert may struggle to see the electrodes clearly due to metal artefacts affecting the image.

One non-radiological intraoperative technique for the determination of the electrode array position within the cochlea is the use of Spread of Excitation (SOE). SOE curves can reveal intraoperative indication positioning problems but cannot be used for final diagnoses as the lack of neural repost (NRT) response does not guarantee good performance [13].

The TransImpedance Matrix (TIM) is an electrical recording method based on the electrical field imaging (EFI) method [14,15]. In TIM, the CI stimulates one contact and records the electrical potential decay along the cochlea on all contacts. In general, the decay constants may depend on the electrode type and location, the cochlear anatomy, and tissue properties. As such, there are many variables to be considered when analyzing the results and the amount of variability in the electrical spread that has been observed among subjects. The detection process is based on potential decay in a medium: *V = k·Q/r*, where *V* is the electrical potential, “*k*” is Coulomb’s constant, “*Q*” is the charge, and “*r*” is the distance to the charge. It can be deduced that the distance correlates inversely with the voltage.

TIM analysis may produce an electrode position “image”, although this position is not only related to the electrical nature of the contacts, as the electrode can be shifted, rotated, or mirrored.

## 2. Material and Methods

This research comprised an observational study using eight fresh human temporal bones. Two different insertions of the Cochlear Nucleus Slim modiolar CI532^®^ pre-curved electrode array (Cochlear, Ltd., Sydney, Australia) were performed, using 22 electrodes. The procedures were performed by an expert surgeon, and the results of both groups of insertions were analyzed.

Two insertions were intentionally made as follows:First: Correct insertion (according to the implant specifications, and with two white marks of the electrode array out of the Round window) (*n* = 8).Second: Intentional fold-over of the electrode array (*n* = 8).

The CI532© is a slim perimodiolar electrode with a diameter of 0.4 mm at the tip and 0.5 mm at the base. It comes with a silicon sheath which has two parts: a white handle and a flexible silicon insertion tube. Both parts are separated by a ‘stopper’ to avoid over-insertions.

The standard insertion technique has been previously described by the authors [16]:The loaded sheath is guided into the cochlea until the sheath stopper reaches the opening of the round window.The sheath handle is inserted and stabilized using straight forceps until the sheath stopper is flush with the opening.With the sheath stopper resting, the electrode is slowly advanced with forceps. When the two white markers align, the electrode is fully inserted.In both cases, the whitemarkers of the electrode arrays were visible after a ”pull back” maneuver was performed.The round window approach was used in both implantations.

Intentional folds were made according to previous experience using a poorly positioned white surgical sheath (90° away from the regular axis). We also filled all cochleae with saline before insertion [16].

All temporal bones were scanned (before and after each electrode insertion) on a MSCT (SIEMENS SOMATOM Sensation 64 Multi-Slice CT). The images were acquired from the squamous portion of each temporal bone to the mastoid process without contrast material (120 kV; 140 mAs; pitch factor: 0.8; collimation slice: 0.6 mm). The CT dose index (CTDIvol) was 32.52 mGy and the average dose-length product (DLP) was 234.75 mGy-cm. The reconstruction process was done with ultra-high-resolution reconstruction algorithms (U75u and U90u). Once the electrode array was inserted, all CT data were acquired and exported in the DICOM file format. Once the raw DICOM files were obtained by the CT system, they were processed, and all data related to the implantation were obtained. The image processing steps were as follows. First, we identified the optimal view of the basal turn of the cochlea by identifying the optimal alignment of the basal turn with the thinner available slice; Second, we saved the electrode array view with a thicker slice (3.5 mm) on average mode visualization; Finally, with a thinner slice (0.6 mm), the visualization in minimum mode was set and the bone view was saved.

Before insertion, all cochleae were measured using the cochlear view as defined by Xu et al. The cochlear duct was measured in three sections of the round window (RW) area, i.e., 90° and 180°, with 0° as the reference angle in RW to the cochlea apex, marking the 270° in a perpendicular axis. Next, the cochlear duct diameter was measured for each segment in a parasagittal section. The difference in diameter was computed as the percentual decrease per segment [11].

All scans were evaluated in the DICOM 3 file format to better visualize the scalar position of the different array areas. For this purpose, custom-made software was developed using MATLAB (R2008b, MathWorks, Natick, MA, USA). To measure electrode distance to the inner wall and electrode position, a script was created to extract the inner wall and the electrode array curves and measure the distance between them at each point of the array. Once we had analyzed the electrode position, the system calculated a parametric curve.

On each temporal bone, the intracochlear position index (ICPI) measurement was analyzed after implantation, as previously described. The ICPI indicates the distance of the electrode to the modiolar wall. This measurement is normalized by the electrode, with “0” being the closest position to the modiolus and “1” the closest to the lateral wall [17].

The TIM procedure used in this study may be described as follows. While an electrode is stimulated, the registered electric potential in all the electrode array is recorded. The next electrode, at a closer position, is then selected for stimulation and the next set of potentials are recorded. This process is automatically repeated until the whole electrode array has been stimulated. As the distance from the stimulating electrode increases from recording electrodes, the potential values are expected to decrease. A typical impedance between two consecutive contacts in the CI532© is between 120–190 Ω.

A biphasic square signal was applied to the electrodes, the amplitude of which was set at 200 current levels, corresponding to 648 μA. The voltage rails were fixed at 0–10.4 V. The determinations were performed at the end of the first trailing edge of the biphasic current pulse. The data were recorded in a matrix. The rows defined the target electrode, i.e., where the measurement was taken, and the columns referred to the active electrode, where the stimulus was produced. 

In a perfect insertion, a maximum value will be obtained on the stimulating electrode and a minimum value on the farthest electrode, although great variability can be observed. The electric potential decays exponentially with distance, so the relative position of the electrodes can be correlated with the measured potential in order to deduce the relative position between them. It is important to consider that other parameters like conductivity and cochlear geometry can contribute to this decay [14].

### 2.1. Description of the Trans Impedance Matrix Patterns

#### 2.1.1. Normal Perimodiolar Position

If a current source is placed in a homogeneous medium, the voltage decreases monotonically as a function of distance to the current source. The relation between distance and absolute voltage drop is nonlinear, depending on the dimensionality and the material properties of the homogeneous medium. In this study, we assumed the absolute value of the voltage difference between the point of stimulation and the recording contacts.

The intracochlear potential map, i.e., the peak levels and slopes, showed substantial variability in the temporal bones studied.

Therefore, in the current study, the apical contacts were placed relatively close to each other. It should also be noted that we found a very narrow cochlear duct, so it was to be expected that close to modiolus we would observe an increase in impedance. This finding was also observed in previous studies [14].

The results of the TIM are presented in two ways: a heat map (Figure 1) and a line graph (Figure 2).

#### 2.1.2. Electrode Fold-Over

In cases of fold-over, the electrical distribution showed that several electrode contacts were placed closely together. In TIM, this situation reflected their electrical similarity as they picked up a similar voltage; this explains the presence of the main diagonal and a second diagonal. The intersection between the main and second diagonal represents the pivot electrode of the fold-over.

In cases of fold-over, the global maximum value will be the stimulating electrode, but another local maximum will appear, indicating fold-over (complete bending). It must be noted that fold-overs can be categorized as tip fold-overs and electrode fold-overs. This differentiation is based on how aggressive the fold-over is. When the fold-over is produced on the tip of the electrode (i.e., only 3–5 electrodes are involved), it is called tip fold-over. A complete fold-over occurs when the electrode array folds in the middle area. This has a severe impact on the operation of the cochlear implant and requires that the procedure be repeated (Figure 3A,B).

The proposed system works in patients without neural responses and eliminates the need for intraoperative imaging tests, with the associated benefit of avoiding radiation and decreasing surgical time, especially in simultaneous bilateral implantations.

## 3. Results

Anatomical Analyses of Temporal Bone:

Eight fresh temporal bones were used. The dimensions of all of them are shown in Table 1, where Diameter A was measured from the round window to the external wall passing through the modiolar area and Diameter B perpendicular to this. The cochlear duct diameter was measured at 0°, 90°, and 180°.

The largest cochlear were found in TB4 and TB6. The cochlear duct diameter relationships at different sites (0°, 90°, 180°) were measured and showed significant differences when compared at 0° and 180° in TB4, TB6, TB7, and TB8.

Regular insertion and electrode position:

In the eight insertions, we observed normal TIM patterns, as described previously. The electrode array position was analyzed in terms of the modiolar proximity of each electrode contact, defined as the distance between the electrode contacts and the modiolus of the cochlea, estimated from the CT scan, using the data exported in DICOM 3 file format with a custom-made software developed using MATLAB (R2008b, MathWorks, Natick, MA, USA).

If we correlate the TIM pattern results with the radiological electrode position, we observe that the better the ICPI, the more homogeneous the TIM voltage pattern (Figure 4). An analysis of the transimpedance slopes adjacent to the stimulating electrode versus its estimated electrode, regarding modiolus distance for the CI532©, observed in CT imaging, showed that higher magnitudes of the transimpedance slopes were correlated to closer perimodiolar position in temporal bones (Figure 5). In only two of the regular insertions (TB5 and TB6) was the ICPI correlating somewhat more strongly with high TIM impedances at those electrodes (Table 2).

TFO detection: intentional TFO insertion was performed in all temporal bones. In all eight temporal bones, the TIM pattern showed a “fold over” effect, with a complete correlation with fold-over detection using CT scan (Table 3).

Additionally, we found a correlation, in all cases, with the location of the electrode where the fold-over appeared and the radiological study (Figure 6).

## 4. Discussion

The transimpedance matrix (TIM) is an electrophysiological method based on a technique described in [14]. In TIM, the cochlear implant (CI) stimulates one contact and records the decay of the electrical potential along the cochlea on all contacts. In general, the decay constants may depend on the electrode type and location, the cochlear anatomy, as well as fluids and the tissue properties. As such, there are many variables to consider when analyzing results.

In a “normal” position, intracochlear voltage dropped monotonically as the distance between stimulation and recording contact increased, i.e., both toward the apex and toward the base in peri-modiolar electrodes. We assumed that the absolute value of the voltage difference between the stimulation contact and the recording contact was an adequate metric. If a current source was placed in a homogeneous medium, the voltage decreased monotonically as a function of distance to the current source. The relation between distance and absolute voltage drop was found to be nonlinear, depending on the dimensionality and the material properties of the medium.

In general, the decay values in TIM depend on the electrode type and location, the cochlear anatomy, and tissue properties. We have observed a more stable situation regarding the impedance distribution along the electrode array in cases with better-positioned electrodes. In cases with an increased distance between the contacts and the modiolar wall, increased impedance was observed, mainly in the apical region of the cochlea compared with the basal turn [14,18].

According with our results, the different types of cochlear morphologies must be taken into account for successful CI insertion. A wide range of cochlear morphologies have been reported. Therefore, it seems essential to analyze the cochlear anatomy prior to CI insertion [6,19].

In fold-over cases (complete bending), the global maximum value will be the stimulating electrode, but another local maximum will appear. When the tip folds over, a matrix with a secondary ridge in addition to the primary ridge can be observed. The intersection between the primary and secondary ridges indicates the location of the “pivoting” electrode, i.e., where the electrode array is bent, providing information about how many electrodes are folded. This situation was observed in all temporal bone electrode insertions where the TIM showed major abnormalities in which the greatest electrical spread curves first decreased with increasing distance but then increased to a second peak where several electrode contacts had been placed closely together [16].

The electric potential values decay with distance, so the relative position of the electrodes can be correlated with the measured potential in order to deduce the relative position between them. It is important to consider that other parameters like conductivity and geometry can contribute to this decay. If other factors are not considered, the decay alone indicates the proximity of the other electrodes but not the exact distance between them.

In the current study, taking into account the intracochlear position of the CI532 electrode array, the apical contacts were placed relatively close to each other. In this situation, an increase in impedance was observed, but only when the situation was controlled. In this respect, it should be noted that with a narrow cochlear duct, the EFI curves may be characterized by larger impedance values [18].

Another possibility that must be considered is the scalar translocation of the electrode array to the scala vestibuli [20,21]. Previous perimodiolar electrodes with thicker diameters seemed to have a tendency for scalar shift. This is related to a loss of residual hearing and a decrease in performance [3,4]. This phenomenon was not observed during the regular insertions in this research.

Although questions remain, the cochlear length seemed not to be associated with scalar dislocation in our study. It was observed that with the CI532, the scalar dislocation rates decreased [22,23].

Open circuits must be also considered. When an open circuit was used, the contact was disconnected from the current source and the electrode was not able to produce the pulse. This resulted in “0” V on the non-stimulating electrodes (in some instances, negative values were observed due to internal bias currents in the implant electronics). On the stimulating electrode, the voltage clips to the amplifier voltage rails. Sometimes, a part of the electrode array stayed outside the cochlea (due to a lack of space, ossifications, the cochlear diameter, etc.). Under such circumstances, the affected electrodes are not in contact with the tissue and their behavior is similar to that of open-circuit electrodes. The potential received is maximal when the stimulated electrode is measured, but the proximal electrodes show lower potential values. However, these values were higher than those observed in open-circuit electrodes.

The intracochlear potential maps, containing peak levels and slopes, showed substantial variability in this study. These preliminary findings must not be overinterpreted. In order to make the system easy to use, an automatic method was also designed for clinical use. This method allows the detection of tip fold-overs as well as open-circuit electrodes and non-inserted electrodes (electrodes not in contact with tissue) intraoperatively [24,25,26].

Although other clinical trials have been published, they use the automatic system that was designed based on the algorithm used in this research. However, there are some important points to be kept in mind:(a)Some of these studies did not achieve significantly different results in TIM conditions, and there was considerable variation among subjects; as such, large groups are required to detect differences among groups. On the other hand, in our research, as the conditions were the same for both insertions, the results were more consistent [27,28].(b)In clinical trials, variables such as impedance modification, inner ear fluids, and the surgical effect of the applied cochlear approach will also affect the electrode placement and electrode discrimination [29,30].(c)It is also important to mention that CT scans (even ConeBeamCT) expose patients to x-rays and should be performed only in difficult situations or in cases where there are problems with fitting or hearing.

We can conclude that clinical data must be obtained in order to provide more consistent results. Additionally, other aspects such as the accuracy of the TIM to predict the electrode shape, incorrect contact orientation, or lateral displacement must be also improved. Correlation with neural response telemetry in an intrasubject study may provide more information in clinical trials [31].

### Limitations

Although this research has the largest serie of controlled insertions with a slim perimodiolar electrode array in fresh Temporal Bone, our analysis is limited by the low number of Temporal Bone specimens. We use exclusively CI 532 model, other different CI models have been not used. In this case we consider that perimodiolar electrodes are more related to this kind of studies and novel techniques. This research is based on MS-CT with its inferior spatial resolution compared to FD-CT. Therefore, further studies should take advantage of FD-CT in order to achieve the highest spatial resolution available.

Modifications of the inner ear fluids is an important factor that has not considered in our research, as we have been working in fresh temporal bones specimens. We use saline before the insertion and all cochleas. Intracochlear fibrosis following CI represents a significant limiting factor for the success of CI users. These changes in the local tissue-electrode interface can also modified the TIM results and of course be detected [32].

## 5. Conclusions

Although electrode array position within the cochlea can be determined by radiology imaging methods, the Trans-Impedance Matrix may provide more information, notably for electrode fold-overs. TIM patterns are correlated with the radiological electrode position; in normal perimodiolar electrode position, we observed a more stable impedance distribution along the electrode array. When a tip fold-over occurred, a matrix with a secondary ridge in addition to the primary ridge was observed in all cases.

## Figures and Tables

**Figure 1 jcm-11-03930-f001:**
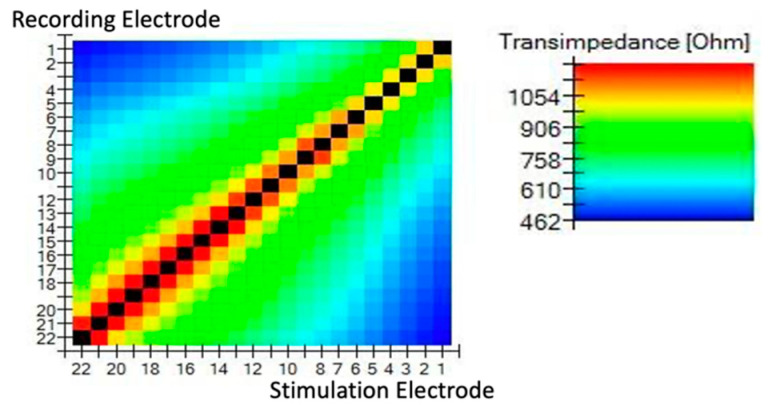
TransImpedance Matrix (TIM) presented on a heat map. In the normal position of the slim perimodiolar electrode, the intracochlear voltage dropped monotonically as the distance between stimulation and recording contact increased, both toward the apex and toward the base.

**Figure 2 jcm-11-03930-f002:**
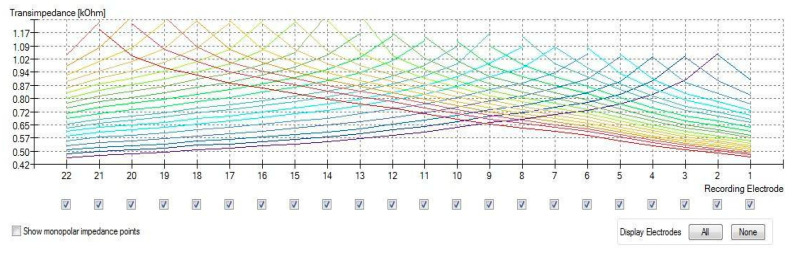
TransImpedance Matrix (TIM) presented in a line graph.

**Figure 3 jcm-11-03930-f003:**
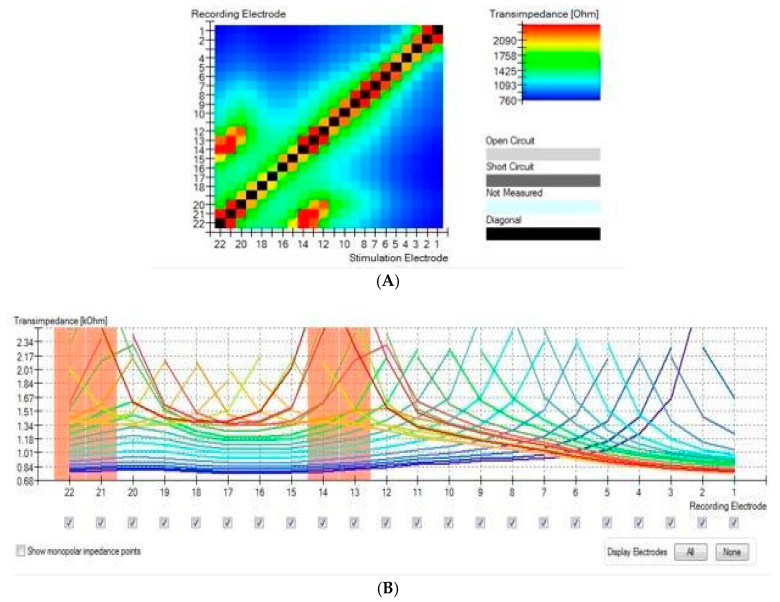
(**A**) Electrode fold-over presented on a heat map. The global maximum value is the stimulating electrode, but another local maximum will appear indicating fold-over. (**B**) Electrode fold-over presented on a line graph.

**Figure 4 jcm-11-03930-f004:**
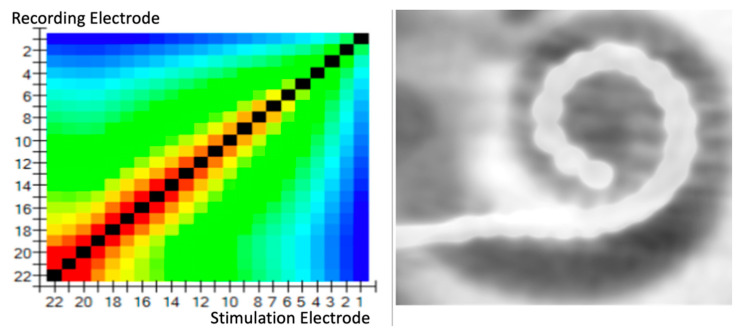
Correlation of TIM patterns with the radiological electrode position; homogeneous TIM.

**Figure 5 jcm-11-03930-f005:**
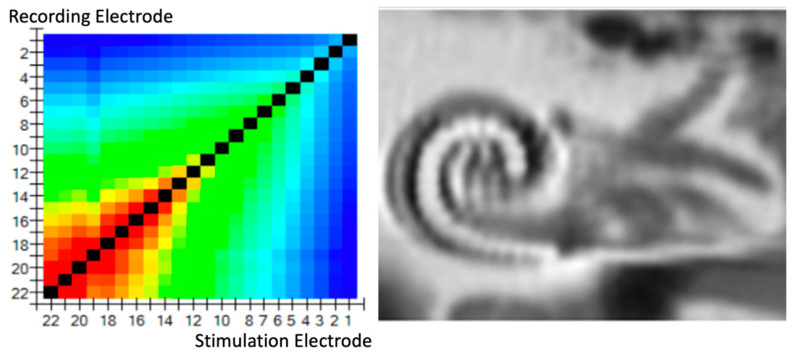
Correlation of TIM patterns with the radiological electrode position. CT imaging shows that higher magnitudes of the transimpedance slopes may be related to closer perimodiolar position in TB.

**Figure 6 jcm-11-03930-f006:**
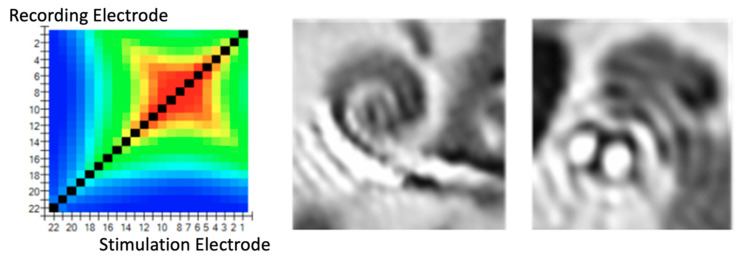
Correlation of TIM patterns with CT imaging shows instances whereby the electrode had folded overs.

**Table 1 jcm-11-03930-t001:** Dimensions of temporal bones (TB). Column A: diameter from round window to the external wall passing through the modiolar area; Column B perpendicular to the previous. The cochlear duct diameter was measured at 0°, 90°, and 180°.

		A	B	0°	90°	180°	Dif 0–180°
TB1	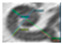	8.45	6.19	1.97	1.83	1.65	16.2%
TB2	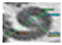	8.54	6.19	2.04	1.59	1.92	5.8%
TB3	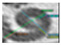	8.11	5.82	1.65	1.43	1.49	9.1%
TB4	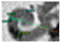	9.58	7.08	2.33	2.20	1.86	20.1%
TB5	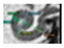	8.85	6.95	2.23	1-92	2.01	9.8%
TB6	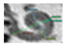	9.28	6.80	2.36	1.92	1.85	21.6%
TB7	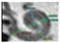	8.20	5.94	2.05	2.17	1.54	24.8%
TB8	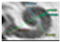	8.59	6.00	2.11	1.81	1.61	23.6%

**Table 2 jcm-11-03930-t002:** Temporal bones (TB), regular insertions. Electrode array position was analyzed in terms of the modiolar proximity of each electrode contact and TIM patterns.

TB1	Regular insertion	Good Perimodiolar position (ICPI = 0.12)	Normal transimpedance slope
TB2	Regular insertion	Good Perimodiolar position (ICPI = 0.19)	Normal transimpedance slope
TB3	Regular insertion	Good Perimodiolar position (ICPI = 0.23)	Normal transimpedance slope
TB4	Regular insertion	Good Perimodiolar position (ICPI = 0.21)	Normal transimpedance slope
TB5	Regular insertion	Mid and apex perimodiloar postion (ICPI = 0.32)	High impedance in tip TIM in apical/mid electrodes
TB6	Regular insertion	Mid and apex perimodiloar postion (ICPI = 0.28)	High impedance in tip TIM in apical/mid electrodes
TB7	Regular insertion	Good Perimodiolar position (ICPI = 0.15)	Normal transimpedance slope
TB8	Regular insertion	Good Perimodiolar position (ICPI = 0.21)	Normal transimpedance slope

**Table 3 jcm-11-03930-t003:** Intentional TFO insertion in all temporal bones. In all TB, the TIM pattern showed a “fold-over” effect, with a complete correlation with fold-over detection using CT scan.

	TFO Electrode	CT Correlation
TFO-TB1	18	Yes
TFO-TB2	17	Yes
TFO-TB3	15	Yes
TFO-TB4	19	Yes
TFO-TB5	10	Yes
TFO-TB6	12	Yes
TFO-TB7	18	Yes
TFO-TB8	17	Yes

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
