# Peer review of "Assessing the Placement of the Cochlear Slim Perimodiolar Electrode Array by Trans Impedance Matrix Analysis: A Temporal Bone Study"

_jcm, 2022, doi:10.3390/jcm11143930_

Round 1

Reviewer 1 Report

The paper appears well constructed and interesting from a scientific point of view, even if the topic is actually well known. The study sample is quite small, but the data are strong.

Maybe, the captions of the tables should be better described.

Finally, at line 114, I do not get the word "Insertion:". Is it a typo, or is it the beginning of a paragraph?

Author Response

Dear reviewer:
Thank you for your comments. I will answer your inputs:
a)The paper appears well constructed and interesting from a scientific point of view, even if the topic is actually well known. The study sample is quite small, but the data are strong.
Comments:
Thanks for the comment. We agree with the reviewer but we consider of interest to have laboratory data of absolutely controlled insertions and perfect knowledge of the intracochlear position using CI stimulation, for further application in human trials.

b)Maybe, the captions of the tables should be better described.
Comments:
Captions of the tables have been improved

c)Finally, at line 114, I do not get the word "Insertion:". Is it a typo, or is it the beginning of a paragraph?
Yes , you are right it is a typo mistake. It has been removed.

Reviewer 2 Report

Dear authors,

compliment to your work and specially for the Discussion. We all agree the study is done on limited number of fresh temporal bone. But the resultrs are important for clinical use. I still think that automatically CT scan (even CB) after surgery is overexpose of x-rays for patients, I think it should be performed only in #difficult# cases or in cases where there are problems at fitting/hearing. Can you address this subject in your discussion?

I do not see the connection with the special issue tittle? vestibular disorders?

Author Response

Dear reviewer:

Thank you for your comments. I will answer your inputs:

  1. a) I still think that automatically CT scan (even CB) after surgery is overexpose of x-rays for patients, I think it should be performed only in #difficult# cases or in cases where there are problems at fitting/hearing. Can you address this subject in your discussion?

Thank you for tour comment, of course we have added this idea in the Discussion.

b)I do not see the connection with the special issue tittle? vestibular disorders?

We submit the manuscript for JCM to be evaluated by editors for different possibilities, and it was considered in that issue due to several other researches with cochlea/vestibular implant. But you are right it can be also in CI issue.